# Extracellular Vesicles from Human Advanced-Stage Prostate Cancer Cells Modify the Inflammatory Response of Microenvironment-Residing Cells

**DOI:** 10.3390/cancers11091276

**Published:** 2019-08-30

**Authors:** Letizia Mezzasoma, Egidia Costanzi, Paolo Scarpelli, Vincenzo Nicola Talesa, Ilaria Bellezza

**Affiliations:** Department of Experimental Medicine, University of Perugia, 06123 Perugia, Italy

**Keywords:** prostate cancer, extracellular vesicles, tumour associated macrophages, IL-1β, caspase-1, ERK1/2, cathepsin B

## Abstract

Prostate cancer (PCa) progression is strictly associated with microenvironmental conditions, which can be modified by cancer-released extracellular vesicles (EVs), important mediators of cell-cell communication. However, the role of EVs in the inflammatory cross-talk between cancer cells and microenvironment-residing cells remains largely unknown. To evaluate the role of EVs in the tumour microenvironment, we treated the non-cancerous prostate cell line PNT2 with EVs isolated from advanced-stage prostate cancer PC3 (PC3-EVs). Caspase-1-mediated IL-1β maturation was evaluated after 24 h incubation with EVs. Moreover, the effect of PC3-EVs on differentiated macrophagic THP-1 cells was assessed by analyzing cytokine expression and PC3 cells migration and proliferation profiles. We illustrated that PC3 cells contain active NLRP3-inflammasome cascade and secrete IL-1β. PC3-EVs affect the PNT2 inflammatory response, inducing caspase-1-mediated IL-1β maturation via ERK1/2-mediated lysosomal destabilization and cathepsin B activation. We also verified that PC3-EVs induce a functional TAM-like polarization in differentiated THP-1 cells. Our results demonstrated that cancer-derived EVs induce an inflammatory response in non-cancerous prostate cells, while inducing an immunomodulatory phenotype in immune cells. These apparently contradictory effects are both committed to strengthening the tumour-promoting microenvironment

## 1. Introduction

Prostate cancer (PCa) is the leading cause of cancer-related death in men in Western countries. Several pathogenic factors have an implication in PCa, including age, diet, hereditary status, and inflammation [1]. The progression of all tumours, including prostate cancer, depends on cell-cell communication between cancer cells and the surrounding microenvironment, which drives cancer progression, immune modulation and the metastatic process. Both cancer and non-cancerous cells cooperate to determine the features of the tumour microenvironment. In the tumour microenvironment, the extracellular matrix is assembled as a three-dimensional supramolecular structure which supplies an active support to both cancer and stromal cells. These latter cells include fibroblasts, mesenchymal cells and immune cells [2,3]. Monocyte-macrophages, the main infiltrating immune cells found in a solid tumour microenvironment, differentiate into tumour-associated macrophages (TAMs). TAMs drive tumour progression by secreting anti-inflammatory and immune-modulatory molecules such as IL-10, thus reducing the immune response against cancer cells [4]. On the other hand, tumour cells themselves secrete several pro-inflammatory cytokines, including interleukin (IL)-6 and IL-1β, strongly implicated in PCa progression. [5,6]. It has been proved that IL-6 is one of the leading cytokines in the progression towards castration resistance [6]. On the other hand, depending on its local doses, the role of IL-1β in cancer is still puzzling [5,7]. In fact, in the presence of low local doses, IL-1β induces an efficient anti-tumour immunity and suppresses tumour formation, whereas in the presence of high local doses, IL-1β is positively related to all the phases of malignancy [5,7,8,9]. It has been shown that in both autocrinal and paracrinal manners, IL-1β stimulates its own production by microenvironment-residing non-cancerous cells; thus promoting the auto-perpetuating inflammatory loop involved in cancer progression [5,9]. IL-1β has been linked to progression and metastasis of many malignancies, such as melanoma, lung, breast and gastric cancers [10,11,12,13,14]. 

In PCa, high immunostaining of IL-1β has been associated with recurrence [15], and high IL-1β levels have been found in tumours [16] and sera [17] of patients with advanced disease. On these bases, Anakirna, an IL-1β receptor antagonist approved in the treatment of systemic inflammatory disorders, has been proposed as a therapeutic strategy for prostate cancer [18]. 

IL-1β secretion is a two-step process requiring the activation of both NF-κB and NLRP3 (nucleotide-binding oligomerization domain (NOD)-like receptor pyrin domain-containing 3) inflammasome-Caspase-1 platform. NF-κB activation by inflammatory stimuli induces biologically inactive pro-IL-1β production which must be proteolytically cleaved, by inflammasome-activated caspase-1 [19]. Different models for NLRP3 inflammasome activation have been proposed: (i) K^+^ efflux driven by ATP-mediated opening of the P2X_7_ purinergic receptor; (ii) peroxynitrite generation, driven by an increase in reactive oxygen/nitrogen species (ROS/RNS); (iii) cathepsin B release into the cytoplasm due to lysosomal destabilisation [19,20,21]. In addition, NF-κB and extracellular signal–regulated kinase 1/2 (ERK1/2)-dependent NLPR3 inflammasome activation has also been reported [22,23]. NLRP3 inappropriate activation has been linked to initiation and progression of a wide range of diseases, including cancer [15,23,24]. Accordingly, advanced-stage melanoma cells possess a constitutively active inflammasome cascade, possibly explaining the high levels of IL-1β found in the melanoma microenvironment [10]. However, NLRP3-inflammasome activation in PCa remains poorly understood.

In addition to inflammatory mediators, the complexity of tumour microenvironment is further enhanced by the presence of extracellular vesicles (EVs) which are membrane-enclosed spherical particles involved in cell-cell communication [25]. EVs contain several functional molecules such as proteins, mRNAs and miRNAs. Upon internalization into target cells, EVs release their cargo which alters the transcriptome and signaling activity of the recipient cells, thus inducing specific phenotypic changes [25]. The prostate gland secretes a specific kind of EV historically called prostasomes, which are physiologically involved in male fertility [26,27]. PCa cells maintain the ability to secrete EVs which may influence the tumour microenvironment [28,29]. Very recently, Ciardello and coworkers [30] demonstrated that EVs secreted by a DU145 prostate cancer subline (DU145R80) transport considerable amounts of key molecules implicated in PCa progression.

With the knowledge that IL-1β, NLRP3-inflammasome and EVs have a pivotal role in the establishment of tumour microenvironment, the contribution of EVs released from advanced-stage PCa cells on the behaviour of microenvironment-residing or infiltrating non-cancerous cells has never been investigated. Therefore, the aim of our study is to understand whether EVs, released from advanced-stage prostate cancer cells, could alter the immune/inflammatory responses of microenvironment-residing cells in a tumour-promoting fashion.

Our results demonstrate that PC3-derived EVs can have a different effect on the microenvironment-residing cells. PC3-EVs induce a TAM-like polarization in THP-1 macrophages and efficiently trigger caspase-1 activation in normal prostate PNT2 cells.

## 2. Materials and Methods

### 2.1. Reagents

All the reagents, unless otherwise stated, were from Sigma-Aldrich (Milan, Italy). ERK 1/2 inhibitor U0126 was obtained from Santa Cruz Biotechnology, Inc. (Dallas, TX, USA), and a 20 mM stock solution in DMSO was prepared. All the antibodies, unless otherwise stated, were from Cell Signaling Technology (Danvers, MA, USA).

### 2.2. Cell Culture and Drug Treatments

Human androgen-independent prostate cancer PC3 cells, human normal prostatic epithelial cells PNT2 and the THP-1 monocytic cell line were obtained from the American Type Culture Collection (ATCC, Manassas, VA, USA). Cell lines were routinely maintained at 37 °C in 5% CO_2_ in RPMI 1640 supplemented with 10% heat inactivated FBS, L-glutamine, 100 units/mL of penicillin and 0.1 mg/mL of streptomycin (Invitrogen, Paisley, UK). Cells were seeded at a density of 50 × 10^3^ cells/cm^2^ and incubated for 24 h before treatments. Cells were treated with 10 µg/mL LPS and after 10 min, exposed to 5 mM ATP [22,23] in the presence or absence of 100 µg/mL of PC3-derived extracellular vesicles (PC3-EVs) for various times (Figure 1A). In independent experiments, 50 mM KCl was added to the culture medium 15 min prior to PC3-EV exposure. KCl treatment did not affect cell viability, ruling out any effect of hypertonicity. In independent experiments, 10 µM U0126 was added to cells 1 h before treatments. DMSO produced no significant toxicity. Control cells with DMSO did not show any significant difference with respect to control cells in RPMI 1640 medium; therefore, all the relative treatments were compared to these latter controls. 

At the end of the treatments, total cell lysates were prepared using RIPA buffer with protease and phosphatase inhibitors. 

THP-1 cells were seeded at 1 × 10^6^ cells in 2 mL of medium in 6-well plates and exposed for 3 days to 300 nM 12-O-Tetradecanoilforbol-13-acetato (TPA) and then exposed to 100 µg/mL PC3-EVs (Figure 1A). In independent experiments, differentiated THP-1 cells were treated with 20 ng/mL hIL-4 for 6 h.

### 2.3. Extracellular Vesicle Isolation

To isolate extracellular vesicles, PC3 cells were grown in RPMI 1640 supplemented with extracellular vesicle-depleted FBS obtained by overnight centrifugation at 100,000× *g*. Extracellular vesicles from PC3 cell culture were obtained after a 3-day culture from culture medium by sequential centrifugation steps [31]. Briefly, PC3 cell medium was pooled and centrifuged at 800× *g* for 10 min to remove intact cells, followed by centrifugation at 2000× *g* for 20 min to remove cell debris. The resulting supernatant was ultracentrifuged at 100,000× *g* for 2 h in an Optima TLX ultracentrifuge with a 60 Ti rotor (Beckman Coulter, Brea, CA, USA). The pellets containing extracellular vesicles were resuspended in PBS supplemented with 1% penicillin/streptomycin solution. Protein concentration was evaluated by measuring absorbance at 280 nm. Extracellular vesicles were stored at −80 °C until use.

EVs were stained for 30 min with 50 µM 1,1′-Dioctadecyl-3,3,3′,3′-Tetramethylindodicarbocyanine, 4-Chlorobenzenesulfonate Salt (DiD′; Thermofisher Carlsbad, CA, USA) and then subjected to another ultracentrifugation step at 100,000× *g* for 2 h. The pellet was resuspended at 1 mg/mL in PBS supplemented with 1% penicillin/streptomycin solution.

### 2.4. Cell viability and Scratch Assay

Cell viability was assessed by the conventional MTT (3-[4,5-dimethylthiazol-2-yl]-2,5-dephenyl tetrazolium bromide) reduction assay [32]. Results were expressed as the percentages of reduced MTT, assuming the absorbance of control cells as 100%.

Wound scratch assay was used to assess the migratory ability of PC3 cells in vitro: 6 × 10^5^ cells seeded in a 24-well dish for 24 h; then, the wound was made with a sterile p200 pipette tip and markers were made to allow observation of cells at the same point. The cells were then rinsed with PBS and grown in RPMI 1640 at 37 °C. Images of the wounds were acquired under a phase-contrast microscope (Leica DM6000B, Milan, Italy) with a digital camera system. The experiments were performed in triplicate and repeated at least three times.

### 2.5. Western Blot Analysis 

Total proteins (20 µg) were separated by 12% sodium dodecyl sulfate-polyacrylamide gel electrophoresis (SDS-PAGE) and were transferred to nitrocellulose membrane. Non-specific binding sites were blocked in Roti-Block (Roth GmbH, Karlsruhe, Germany) for 1 h at room temperature. The membranes were blotted overnight at 4 °C with the following anti-human Abs diluted in Roti-Block: anti-NLRP3(D4D8T) mouse monoclonal antibody (mAb) (#15101), anti-Caspase-1 rabbit polyclonal antibody (pAb) (#2225), anti-Phospho-NF-kB p65 (Ser536) (93H1) rabbit mAb (#3033), anti-Phospho-p44/42 MAPK (ERK 1/2) (Thr202/Tyr204) rabbit pAb (#9101), anti-p44/42 MAPK (ERK 1/2) rabbit pAb (#4695), anti-IL-1β (3A6) mouse mAb (#12242), pAb anti-Cathepsin B (#MA5-32651) (ThermoFisher) (1:1000 dilution). After washing with TBST, blots were incubated for 1 h at room temperature with the appropriate HRP-conjugated secondary Abs (1:2000 dilution) and revealed using the enhanced chemi-luminescence (ECL) system (Amersham Pharmacia Biotech, Milan, Italy). Membranes were stripped and re-probed with anti-β-actin mAb (I-19) antibody (1:400) (Santa Cruz Biothecnology) as a loading control. Densitometric analyses were performed with ImageJ software (https://imagej.nih.gov/ij/).

### 2.6. ROS Generation

The 2’,7’-dichlorodihydrofluorescein diacetate (DCFH-DA) method was used to detect intracellular ROS levels [32].

The fluorescence of 2’,7’-dichlorofluorescein was detected at 485 nm excitation and at 535 nm emission using a Titertek Fluoroscan II (Flow Laboratories, McLean, VA, USA). Results were expressed as % of the control DCF fluorescence.

### 2.7. Measurements of Secreted IL-1β 

Measurements of secreted IL-1β were performed in 100 µL supernatant collected after 48 h culture. Human IL-1β levels were determined by the specific ELISA kit, according to the manufacturer’s guidelines (eBioscience, San Diego, CA, USA) [33].

### 2.8. Real Time PCR 

Total RNA was isolated with TRIZOL Reagent (Invitrogen,) according to the manufacturer’s instructions and cDNAwas synthesized using iScript cDNA synthesis kit (Bio-Rad Lab, Hercules, CA, USA). Real-time PCR was performed using the iCycleriQ detection system (Bio-Rad Lab) and SYBR Green chemistry as previously described [34]. The n-fold differential ratio was expressed as 2^−ΔΔCt^. A list of primers is reported in Table 1.

### 2.9. Fluorescence Microscopy Analyses

PNT2 cells, seeded on glass coverslips, were treated for 24 h with 100 µg/mL PC3-EVs. Lysosomes were stained with 50 nM Lysotracker-green (Life Technologies) and nuclei were stained with 0.1 mg/mL Hoechst (Life Technologies) [34]. 

Differentiated THP-1 cells, seeded on glass coverslips, were exposed to 100 µg/mL PC3-EVs previously stained with DiD. At the indicated time points, cells were fixed with 4% PFA for 20 min at room temperature and F-actin was stained with fluorescein isothiocyanate (FITC)-labelled phalloidin (1:250) for 30 min at room temperature and cell nuclei were counterstained with 4’, 6’-diamidino-2phenylindole (DAPI).

Cells were then rinsed in PBS, mounted and analysed with a Zeiss Axio Observer Z1 equipped with Apotome and digital Camera Axiocam MRm (Zeiss, Oberkochen, Germany).

### 2.10. Statistical Analysis

Results were expressed as means ± SD of at least three independent experiments performed at least in quadruplicate using GraphPad-PRISM software. Student’s T-test was used to compare two groups. A *p* value less than 0.05 was considered significant. 

## 3. Results 

### 3.1. PC3-Secreted EVs Induced TAM-Like Polarization in THP-1 Differentiated Macrophages

It has already been established that prostate cancer cell lines secrete extracellular vesicles (EVs), known as prostasomes, similar to that produced by the prostate gland [29,31]. 

Based on the knowledge that prostasomes from human semen can have immunosuppressive properties [35], we asked whether EVs isolated from PC3 cells might affect macrophage polarization. THP1 cells were differentiated in M0 macrophages by a 300 nM TPA 3-day exposure and then treated for another 6 h with 100 µg/mL PC3-derived EVs (PC3-EVs) (Figure 1A). We found that in differentiated THP1 cells, the exposure to PC3-EVs induces an increase in IL-10 expression without affecting IL-12 expression (Figure 1B). This gene expression profile resembles the characteristic of M2/TAM macrophages, i.e. IL-12^low^/IL-10^high^. Hence, the extent of IL-10 gene expression induced by PC3-EVs is comparable to that induced by 20 ng/mL IL-4, the classical M2 inducer (Figure 1B). We also found that PC3-EVs adhere to differentiated THP-1 cell membrane after a 15-min incubation and can be detected in the intracellular compartment after 1 h (Figure 1C). It is conceivable that, by releasing their cargo, PC3-EVs can drive a behavioural change in the recipient cell.

It is known that the tumour microenvironment can facilitate tumour cell growth by inducing TAM polarization in peritumoural macrophages [4]. To understand whether THP-1 cells exposed to PC3-EVs could favour tumour progression, we exposed PC3 cells to the conditioned medium of differentiated THP-1 cells, previously treated with PC3-EVs (pCM) (Figure 1A). We found that pCM induces a significant increase in PC3 cell proliferation (Figure 1D). We then analysed the effects of pCM on PC3 cell migration by scratch assay. As shown in Figure 1E, PC3 cells exposed to pCM have a higher migration ability compared to cells exposed to M0 differentiated THP-1 conditioned medium. These data indicate that EVs-induced change in the THP-1 phenotype can render macrophages capable of stimulating the prostate cancer cells’ malignant phenotype.

### 3.2. IL-1β-Secreting PC3 Cells Contain an Active NLRP3-Inflammasome Cascade

Besides the induction of TAM polarization in peritumoural macrophages, the tumour microenvironment can facilitate tumour cell growth by inducing an IL-1β-dependent auto-inflammatory loop involving non-cancerous cells [5]. Therefore, we first determined the features of the cancerous (PC3) and non-cancerous (PNT2) prostate cells focusing on the inflammasome cascade. The supernatants from androgen-independent PC3 cells exhibit IL-1β levels that are 40% higher than normal prostate epithelial PNT2 cells (Figure 2A). The result is in agreement with the high IL-1β levels found in tissue and serum of Pca patients with advanced disease [16,17]. Compared to PNT2 cells, PC3 cells exhibit a higher active NF-κB, required for IL-1β gene expression (Figure 2B). The high levels of active NF-κB seem to be a feature of androgen receptor negative cells. Indeed, androgen receptor-expressing cells, LNCaP and MDAPCa2b, show negligible nuclear accumulation of NF-κB (Appendix A). Accordingly, PC3 cells show higher IL-1β mRNA expression (Figure 2C) that mirrors the pro-IL-1β protein expression (Figure 2D). Compared to PNT2 cells, PC3 cells display a higher expression of NLRP3 and cleaved caspase-1, essential to pro-IL-1β maturation (Figure 2E). These data suggest that advanced stage prostate cancer cells acquire the ability to secrete active IL-1β by increasing inflammasome-cascade components.

### 3.3. PC3-Derived EVs Induce Caspase-1 Activation and IL-1β Maturation in PNT2 Cells

Since EVs secreted by PC3 cells may contribute to the auto-perpetuating inflammatory loop typical of a tumour microenvironment [5], we exposed PNT2 cells to 100 µg/mL of PC3-derived EVs (PC3-EVs) for 24 h with or without 10 µg/mL LPS + 5 mM ATP as inflammatory stimulus. PC3-EV exposure doubles the NLRP3 expression, causes caspase-1 cleavage (Figure 3A) and IL-1β maturation, independent of LPS + ATP (Figure 3B). Under these experimental conditions, LPS plus ATP treatment fails to cleave caspase-1 and to mature IL-1β (Figure 3A,B). PC3-EV treatment also induces an increase in IL-1β protein expression (Figure 2B). By analyzing NF-κB p65 phosphorylation, we found that a 3-h PC3-EV exposure induces a slight NF-κB activation (Figure 3C), followed by a slight but significant increase in IL-1β mRNA expression after a 6-h PC3-EV exposure (Figure 3D). On the other hand, EVs isolated from normal PNT2 cells fail to induce caspase-1 cleavage (Figure 3E). These data indicate that prostate cancer-derived EVs are capable of inducing inflammasome activation in non-cancerous cells.

### 3.4. PC3-Derived EVs Activate Caspase-1 via ERK 1/2

To determine the molecular mechanism responsible for PC3-EVs-mediated inflammasome activation, we analysed the activation of ERK1/2, a mitogen activated protein kinase already connected to the inflammasome activation [22,23]. A 24 h exposure to 100 µg/mL PC3-EVs causes a consistent activation of ERK 1/2 in PNT2 cells (Figure 4A). We then exposed PNT2 cells to 10 µM U0126, a specific MEK1/2 inhibitor known to act on ERK1/2 phosphorylation/activation, in the presence or in the absence of 100 µg/mL PC3-EVs. The exposure to U0132 abolishes caspase-1 cleavage (Figure 4B) and PC3-EVs-induced IL-1β maturation (Figure 4C), thus indicating that PC3-EVs induce inflammasome activation via ERK1/2.

### 3.5. PC3-Derived EVs Trigger Cathepsin B Activation and Lysosomal Destabilization via ERK 1/2 

To investigate whether EVs-induced effects rely on other molecular pathways acknowledged to be involved in NLRP3-inflammasome activation [20], we analyzed the effects of PC3-EVs on K^+^ efflux, ROS generation and cathepsin B activation. We first inhibited K^+^ efflux by exposing PNT2 cells to 50 mM KCl in the presence or in the absence of 100 µg/mL PC3-EVs and found that KCl does not affect EV-induced caspase-1 cleavage (Figure 5A). To investigate the effects of EVs on ROS generation, we exposed PNT2 cells to 100 µg/mL PC3-EVs for 24 h. We found that PC3-EV exposure does not affect ROS generation (Figure 5B), thus excluding the involvement of both K^+^ efflux and ROS generation in EV-induced effects. It is worth mentioning that a 24 h exposure to PC3-EVs increases PNT2 cell viability to 178.17 ± 26.67% of the control value. 

Another mechanism implicated in inflammasome activation relies on cathepsin B activation [19,20]. We found that a 24 h PC3-EV exposure induces an intense increase in the cathepsin B cleaved/active form (Figure 5C), suggesting an involvement of this enzyme in EV-induced effects. To determine a potential link between ERK 1/2 and cathepsin B activation, we pretreated PNT2 cells with 10 µM U0126, prior to 24 h PC3-EV exposure and found that ERK1/2 inhibition reduces both basal and PC3-EVs-induced cathepsin B cleavage (Figure 5D). 

Lysosomal destabilisation is a crucial event leading to caspase-1 activation via cathepsin B [20]. By using lysotracker green staining, we found that a 24 h exposure of PNT2 cells to PC3-EVs causes a lysosome enlargement (Figure 5E), indicative of lysosomal destabilization. To further confirm the link between ERK 1/2 activation and lysosomal destabilization, we pretreated PNT2 cells with U0126 before a 24 h PC3-EV exposure. Pharmacological inhibition of ERK1/2 reduces the lysosome enlargement caused by EV exposure (Figure 5E). Thus PC3-derived EVs, by inducing ERK1/2 phosphorylation, may efficiently trigger caspase-1 activation via lysosomal destabilization and cathepsin B activation.

## 4. Discussion 

This is the first report that illustrates that extracellular vesicles (EVs) secreted by advanced stage prostate cancer (PCa) cells induce an inflammatory phenotype in non-cancerous prostate cells and a TAM-like polarization in immune cells. By comparing PCa cells to non-cancerous prostate epithelial PNT2 cells, we also documented a higher IL-1β secretion related to basal activation of NF-κB and NLRP3/caspase1 pathway in advanced stage prostate cancer cells. 

It is well acknowledged that modifications to the microenvironment are deeply involved both in cancer development and progression [3]. Cancer cells reside in a microenvironment that includes endothelial cells, fibroblasts, different immune cells and non-cancerous cells. Each of these cells contribute to the microenvironment features via the secretion of both soluble factors and EVs that cooperate in sustaining tumour growth [15]. It has been demonstrated that EVs, released from several types of cancer cells, induce macrophage polarization towards a tumour-associated macrophage (TAM) phenotype [36,37,38]. TAMs are a tumour-associated immune cell population involved in cancer progression [4]. In this report we have shown that prostate cancer-derived EVs induce an immunosuppressive TAM-like phenotype in M0 macrophages. In fact, upon exposure to PC3-EVs, the differentiated THP-1 cells express low levels of the pro-inflammatory cytokine IL-12 and high levels of the anti-inflammatory cytokine IL-10, the cytokine signature characteristic of TAM-like macrophages [39]. This result correlates with the notion that one of the roles of EVs released by the prostate gland is the inhibition of the female immune system to guarantee a successful reproductive cycle [35]. We can suppose that EV effects depend on uptake by the THP-1 cells. In fact, already after 15 min, EVs adhere to the cell membrane and after 1 h EVs can be found in the cytoplasm of THP-1 cells. Therefore, after the uptake, EVs may fuse with the endosomal membrane to liberate their content [40] that, in turn, can induce a specific phenotypic change in the recipient cell. It has recently been reported that EVs from prostate cancer cells can transfer αvβ6 integrin to monocyte, thus inducing an M2-like polarization [36]. Several routes for EVs uptake by recipient cells have been hypothesised [40]. These include phagocytosis, clathrin- and caveolin-mediated endocytosis and the fusion with the plasma membrane [40]. Our results exclude the latter mechanism of uptake since spherical particles can be observed in the cytoplasm of THP-1 cells upon EVs treatment. Our in vitro settings can mimic the conditions found in vivo. As a matter of fact, EVs have been identified both extracellularly, in the interstitial tissues, as well as in the cytoplasm of the metastatic prostate cancer cells [29]. 

One of the main functions of TAMs is to support oncogenesis and disease progression by promoting tumour-protective adaptive immunity and by influencing tumour growth [4,41]. Several molecules, including IL-10 [42], IL-32 [43], epidermal growth factor [44] and growth arrest-specific gene 6 [45], have been associated to the tumour-promoting function of TAMs. We have demonstrated that PC3-EV-induced TAM-like macrophages are capable of increasing cancer cell proliferation and migration, thus supporting the idea that EVs might exert a tumour-promoting function. 

However, since EVs are a means of communication between the cell of origin and other microenvironment-residing cells [15], their effect can depend on the recipient cell. Data in the literature show that ovarian cancer-derived EVs activate normal fibroblast into a cancer-associated fibroblast phenotype, capable of sustaining tumour growth [46]. Moreover, it has been shown that prostate cancer-derived EVs can drive epithelial-mesenchimal transition in normal prostate cells [47]. It has also been reported that exosomes, isolated from the murine prostate cancer cell line TRAMP-C1, are involved in tumour and bone cell interactions [48]. These data indicate that cancer-derived EVs can affect non-cancerous cell behaviour. 

It should be also kept in mind that inflammation plays a major role in PCa initiation and progression. Stimuli such as urine reflux, uric acid crystals, and the prostate dysbiotic microbiome can indeed cause inflammation within the prostate, prompting prostate cancer [49]. IL-1β, one of the most abundant pro-inflammatory cytokines of the tumour microenvironment, and NLRP3-inflammasome activation have been involved in tumourigenesis and progression of PCa [1,49,50]. We have demonstrated that PC3-derived EVs (PC3-EVs), independently of a pro-inflammatory milieau, induce a pro-inflammatory response in non-cancerous PNT2 cells. The pro-inflammatory response is driven by an increased expression of NLRP3, caspase-1 activation and IL-1β maturation. PNT2-derived EVs have not activated caspase-1, indicating that the observed effects are characteristic of PC3-derived EVs. Via activation of NLRP3 inflammasome, PC3-derived EVs are likely to contribute to the auto-perpetuating inflammatory loop, a tumour characteristic [5]. This hypothesis is strengthened by our findings that PC3 cells contain active NLRP3-inflammasome and secrete high IL-1β levels. IL-1β, together with NF-κB, has already been associated with the progression of PCa [1,50,51,52,53]. NF-κB activation in cancer cells leads to the amplification of the inflammatory response, sustaining the production of molecular mediators in the carcinogenic process [53]. Moreover, NF-κB, whose activation increases with the Gleason score [54], has been shown to protect prostate cells from apoptosis, to stimulate proliferation, and to play an important role in the selection of hormone independence [55]. The exposure to PC3-EVs induces, in the recipient cell, the activation of NF-κB, detected as phosphorylated p65, resulting in an increase in the IL-1β gene and protein expression. Thus, we are proposing that PC3-EVs act as a priming signal prompting the activation of NF-κB, as already reported for microvesicles released by fat-laden cells undergoing lipotoxicity [56]. We have shown that PC3-EVs could induce NLRP3 inflammasome and caspase-1 activation. In order to decipher the molecular mechanism underlying PC3-EV-induced effects, we analysed the molecular pathways acknowledged to be related to caspase-1 activation and found that neither K^+^ efflux nor ROS generation is mandatory for PC3-EV-induced effects. The molecular mechanism of PC3-EVs-related caspase-1 activation is instead connected to ERK1/2 signaling. In fact, pharmacological inhibition of ERK1/2 abolishes PC3-EV-induced caspase-1 cleavage and decreases the maturation of IL-1β. These data agree with recent findings that link ERK1/2 to NLRP3-inflammasome cascade activation in human immune cells exposed to pro-inflammatory stimuli [23,57].

Interestingly, PC3-EV exposure induces an increase in cathepsin B cleaved/active form mediated by ERK1/2 activation as recently shown in DSS-induced colitis where inflammasome activation, superimposed by deoxycolic acid, is triggered by ERK1/2 and cathepsin B [58]. Furthermore, PC3-EVs induce an ERK1/2-dependent lysosomal destabilization. This finding is in agreement with reports showing that ERK1/2 is involved in the lysosome destabilization crucial to cathepsin B-induced caspase-1 activation [57].

These results indicate that EVs secreted by cancer cells can alter the non-cancerous counterpart to support tumour growth. However, prostate cancer is a heterogeneous disease comprising indolent tumours and aggressive tumours that metastasise rapidly [59]. Another layer of heterogeneity regards the metastatic site. The most common metastatic sites for prostate cancer are indeed bone, distant lymph nodes, liver, and thorax [60]. Although the metastatic process can depend on cancer cell themselves, the characteristics of the metastatic niche, that allow habitability, can also influence the metastatic process. This concept highlights the need for further research to understand the potential effects of EVs on other microenvironment-residing cells of the metastatic niche, including fibroblasts.

## 5. Conclusions

Collectively, our results demonstrate that PC3-derived EVs can affect, in different ways, microenvironment-residing cells. PC3-EVs induce a TAM-like polarization in THP-1 macrophages and efficiently trigger caspase-1 activation via lysosomal destabilization and cathepsin B activation in normal prostate PNT2 cells. To our knowledge, this is the first study showing a pro-inflammatory role for EVs in cancer, a mechanism that should be explored in other cancerous diseases. These findings, highlighting the role of EVs as master regulators of cell-cell communication in PCa, may contribute to the development of novel microenvironment-targeted cancer therapies for prostate cancer patients.

## Figures and Tables

**Figure 1 cancers-11-01276-f001:**
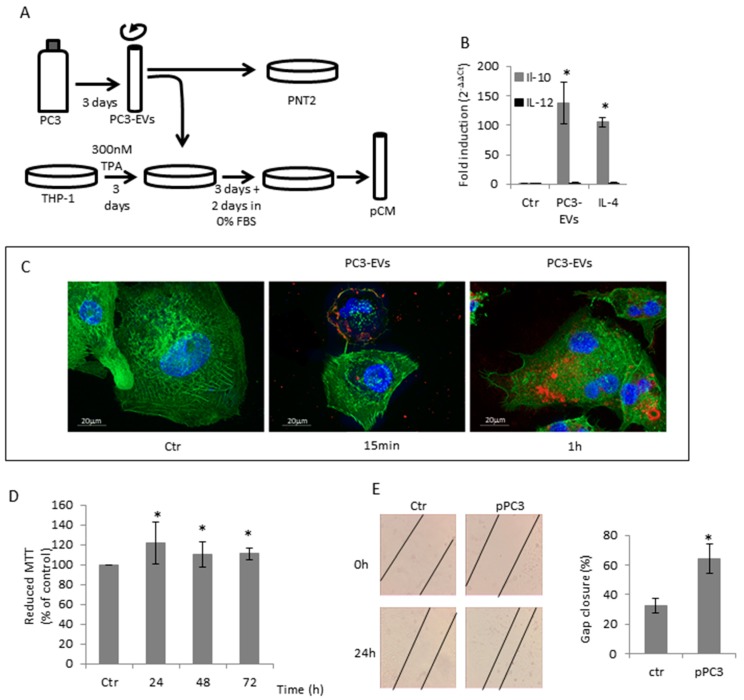
PC3-secreted EVs induced TAM-like polarization in THP-1 differentiated macrophages. (**A**) Diagrammatic representation of treatments. (**B**) TPA-differentiated THP-1 cells were exposed to 100 µg/mL PC3-Evs or 20 ng/mL IL-4 for 6 h and IL-10 and IL-12 gene expression was assessed by qRT-PCR. Gene expression values were normalized to HPRT and presented as 2^−ΔΔCt^. Relative mRNA gene abundance in untreated cells (Ctr) was assumed as 1. Data represent mean ± SD (*n* = 6). * *p* < 0.05 vs. control cells. (**C**) TPA-differentiated THP-1 cells were exposed to 100 µg/mL DiD’-stained PC3-Evs. At the indicated time points, cells were fixed, actin filaments stained with FITC-labelled phalloidin and nuclei were counterstained with DAPI. The images are representative of one out of three separate experiments. Magnification 63×. (**D**) PC3 cells were exposed to pCM for the indicated times and cell viability was determined by MTT assay. MTT reduction in untreated cells (Ctr) was assumed as 100%. Data represent mean ± SD (*n* = 6). * *p* < 0.05 vs. control cells. (**E**) PC3 cells were grown to confluence, scratched and exposed to pCM. The rate of migration was measured by quantifying the distance between the edges of the scratch. The width of the gap at time 0 was considered to be 100% and was used to calculate the % of gap closure (reported in the graph). Data represent the mean ± SD (*n* = 3). * *p* < 0.05 vs. control cells.

**Figure 2 cancers-11-01276-f002:**
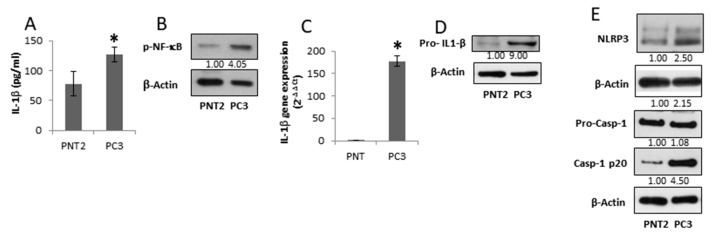
IL-1β-secreting PC3 cells contain an active NLRP3-inflammasome cascade. Normal prostate PNT2 and prostate cancer PC3 cells were grown for 48 h, and (**A**) IL-1β in the supernatant was assessed by ELISA. Results are reported as mean ± S.D. *n* = 5 separate experiments. * *p* < 0.05 vs. PNT2 cells; (**B**) p-NF-κB (p65) expression was evaluated in total cell extract analyzed by Western blotting. Β-actin was used as a loading control. The images are representative of one out of *n* = 3 separate experiments; (**C**) IL-1β gene expression was assessed by qRT-PCR. Gene expression values were normalized to HPRT and presented as 2^−ΔΔCt^. Relative mRNA gene abundance in PNT2 cells was assumed as 1. Data represent mean ± SD (*n* = 5). * *p* < 0.05 vs. PNT2 cells; (**D**) pro-IL-1β, and (**E**) NLRP3, caspase-1 full length and mature caspase-1-p20 expression was evaluated in total extract analyzed by Western blotting. β-actin was used as a loading control. The images are representative of one out of *n* = 3 separate experiments. Numbers below the images represent the ratio between the respective protein and β-actin band intensity.

**Figure 3 cancers-11-01276-f003:**
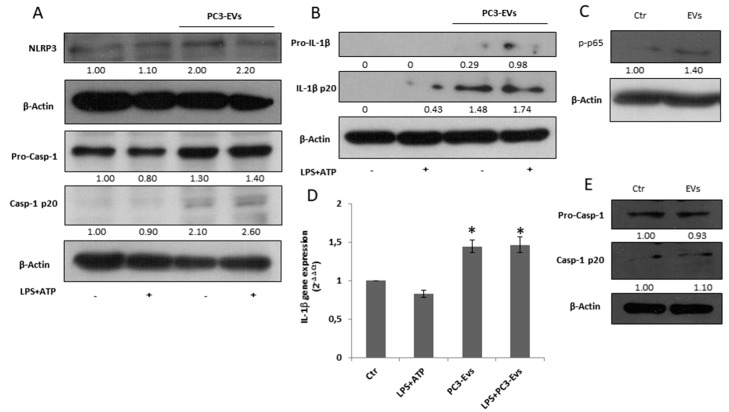
PC3-derived EVs induce caspase-1 activation and IL-1β maturation in PNT2 cells.PNT2 cells were treated with 100 µg/mL of PC3-derived EVs (PC3-EVs) either in the presence or in the absence of 10 µg/mL LPS + 5 mM ATP (LPS + ATP). After a 24 h exposure, (**A**) NLRP3, caspase-1 full length and mature caspase-1-p20 expression and (**B**) pro-IL-1β full length and mature IL-1β-p20 expression were evaluated in total extract analyzed by Western blotting. β-actin was used as a loading control. The images are representative of one out of at least three separate experiments. (**C**) After a 3 h exposure, p-NF-κB (p65) expression was evaluated in total extract and analyzed by Western blotting. β-actin was used as a loading control. The images are representative of one out of at least three separate experiments. (**D**) qRT-PCR of IL-1β gene after a 6 h exposure. Gene expression values were normalized to HPRT and presented as 2^−ΔΔCt^. Relative mRNA gene abundance in PNT2 cells was assumed as 1 (Ctr). Data represent mean ± SD (*n* = 5). * *p* < 0.05 vs. PNT2 cells. (**E**) PNT2 cells were treated with 100 µg/mL of PNT2-derived EVs (PNT2-EVs) for 24 h and caspase-1 full length and mature caspase-1-p20 expression was evaluated in total extract analyzed by Western blotting. β-actin was used as a loading control. The images are representative of one out of *n* = 5 separate experiments. Numbers below the images represent the ratio between the respective protein and β-actin band intensity.

**Figure 4 cancers-11-01276-f004:**
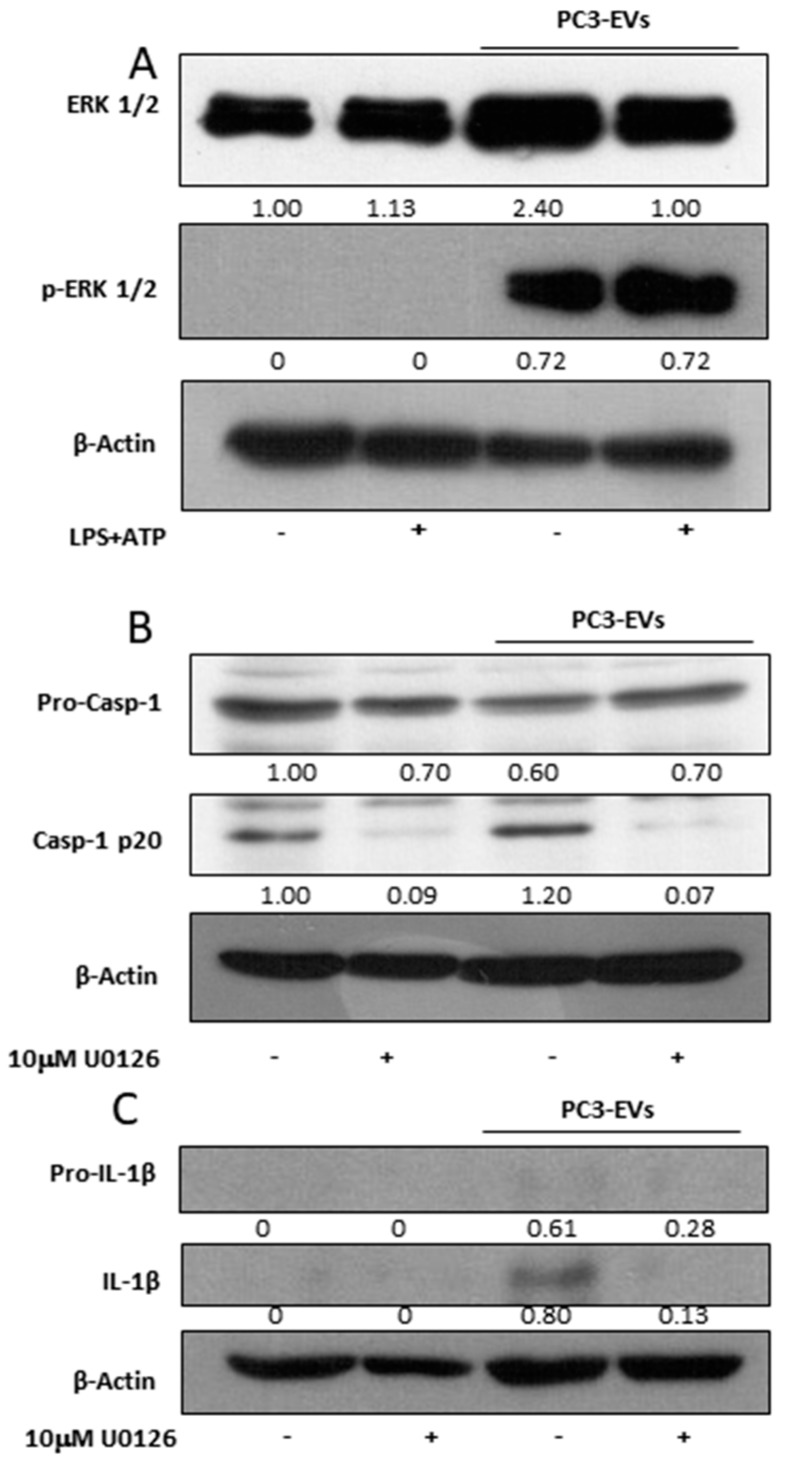
PC3-derived EVs activate caspase-1 via ERK1/2. PNT2 cells were treated with 100 µg/mL of PC3-derived EVs (PC3-EVs) for 24 h (**A**) in the presence or in the absence of 10 µg/mL LPS + 5 mM ATP (LPS + ATP) and the expression of phosphorylated ERK1/2 was analyzed by Western blotting. PNT2 cells were pretreated for 1 h with 10 µM U0126 and then treated for 24 h with 100 µg/mL PC3-EVs. (**B**) caspase-1 full length and mature caspase-1-p20 expression and (**C**) pro-IL-1β full length and mature IL-1β-p20 expression were evaluated in total extract analyzed by Western blotting. β-actin was used as a loading control. The images are representative of one out of *n* = 3 separate experiments. Numbers below the images represent the ratio between the respective protein and β-actin band intensity.

**Figure 5 cancers-11-01276-f005:**
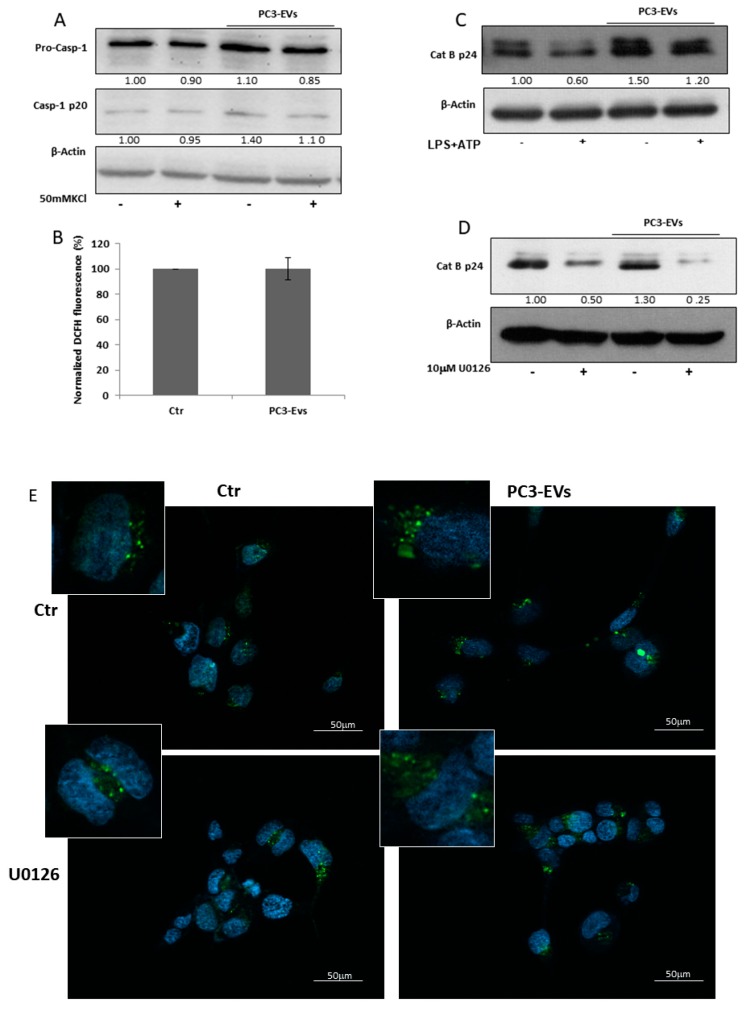
PC3-derived EVs trigger cathepsin B activation and lysosomal destabilization via ERK1/2.PNT2 cells were treated with 100 µg/mL of PC3-derived EVs (PC3-EVs) for 24 h and (**A**) the expression of caspase-1 full length and mature caspase-1-p20 was evaluated in total extracts by Western blotting either in the presence or in the absence of 50 mM KCl; (**B**) ROS levels were assessed by DCFH fluorescence. Results are reported as mean ± SD. Fluorescence of untreated cells, normalized to cell viability, was assumed as 100%; (**C**) the expression of cathepsin B mature p24 form in total extracts was analyzed by Western blotting either in the presence or in the absence of 10 µg/mL LPS + 5 mM ATP (LPS + ATP). The images are representative of one out of at least three separate experiments. PNT2 cells were pretreated for 1 h with 10 µM U0126 and then treated for 24 h with 100 µg/mL PC3-derived EVs (PC3-EVs) and (**D**) the expression of cathepsin B mature p24 form was evaluated in total extracts by Western blotting. β-actin was used as a loading control. The images are representative of one out of *n* = 3 separate experiments. Numbers below the images represent the ratio between respective protein and β-actin band intensity; (**E**) lysosomal compartment, stained with lysotracker green, was analyzed by fluorescence microscopy. Nuclei ware counterstained with Hoechst. Magnification 40 ×. The images are representative of one out of three separate experiments.

**Table 1 cancers-11-01276-t001:** List of Primers.

Gene	Primers Forward (F) and Reverse (R)
IL-1β	**F**: CCAGCTACGAATCTCCGACC; **R**: CATGGCCACAACAACTGACG
IL-12 p40	**F**: CGGTCATCTGCCGCAAA; **R**: TGCCCATTCGCTCCAAGA
IL-10	**F**: CGAGATGCCTTCAGCAGAGT; **R**: CGCCTTGATGTCTGGGTCTT
HPRT	**F**: TGACACTGGCAAAACAATGCA; **R**: GGTCCTTTTCACCAGCAAGCT

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
