# Peer review of "Extracellular Vesicles from Human Advanced-Stage Prostate Cancer Cells Modify the Inflammatory Response of Microenvironment-Residing Cells"

_cancers, 2019, doi:10.3390/cancers11091276_

Round 1

Reviewer 1 Report

Inflammatory response are frequently studied in prostate cancer. They may be relevant to early stages and prostate cancer progression. This paper is important for studies on extracellular vesicles and inflammatory response. Although several results are novel, additional experiments are recommended. The authors should consider some suggestions for improvement:

Fig. 1D, effect on migration of PC3 cells has been described. It is not clear whether this effect could be abolished by a specific antibody? This issue should be addressed by the authors.

Fig. 2, high levels of p-nuclear factor kappa B could be limited to a single cell line (PC3). This could be simply a result of the effect of absence of androgen receptor. Recently, several novel androgen-sensitive cell lines have been used and the authors should consider this fact in additional experiments. 

Fig. 2B, IL-10 and IL-12 are mentioned, however it is not clear from figure legend how they were added. Please clarify. 

Author Response

Reviewer 1

We would first thank the reviewer for the stimulating comments. We have amended the MS according to reviewer’s suggestions and hope that it is now suitable for publication.

-Inflammatory response are frequently studied in prostate cancer. They may be relevant to early stages and prostate cancer progression. This paper is important for studies on extracellular vesicles and inflammatory response. Although several results are novel, additional experiments are recommended. The authors should consider some suggestions for improvement:

-Fig. 1D, effect on migration of PC3 cells has been described. It is not clear whether this effect could be abolished by a specific antibody? This issue should be addressed by the authors.

Response

It is known that tumour associated macrophages can prompt tumour progression. On these bases, we analysed the effect of THP-1 conditioned medium on PC3 cells to demonstrate that PC3-EVs can affect macrophages polarization toward a tumour-promoting phenotype. Several molecules have been involved in the tumour promotion by TAMs. This concept has been added to the text. Pag. 13 lines 393-395

-Fig. 2, high levels of p-nuclear factor kappa B could be limited to a single cell line (PC3). This could be simply a result of the effect of absence of androgen receptor. Recently, several novel androgen-sensitive cell lines have been used and the authors should consider this fact in additional experiments.

Response

We added a supplementary fig. 1 showing NF-κB nuclear localization in androgen sensitive and independent cell lines. Our results are in agreement with the work of Gasparian and collaborators, that now appear in the reference list. Pag. 7-8 lines 245-247. Moreover, the implication of NF-κB has been discussed in pag. 14 lines 422-425.

-Fig. 2B, IL-10 and IL-12 are mentioned, however it is not clear from figure legend how they were added. Please clarify. 

Response

We apologise with the reviewer for the poor clarity of the sentences. We didn’t add IL-12 and IL-10 to the culture medium but analysed their gene expression in THP-1 cells exposed to EVs. In an attempt of improving clearness we modified both the results section and the legend to fig. 1 as follows:

Results section pag 5, lines 179-182 now reads: “We found that in differentiated THP1 cells the exposure to PC3-EVs induces an increase in IL-10 expression without affecting IL-12 expression (Fig. 1B). This gene expression profile resembles the characteristic of M2/TAM machrophages, i.e. IL-12low/IL-10high.” Pag. 5 lines 201-203.

Legend to figure 1 now reads: “TPA-differentiated THP-1 cells were exposed to 100µg/ml PC3-EVs or 20ng/ml IL-4 for 6h and IL-10 and IL-12 gene expression were assessed by qRT-PCR.” pag 7, lines 223-224.

Reviewer 2 Report

The authors describe the role of prostate cancer EV's in their micro-environment (and macrophages) very nicely. The role of IL-1B, Caspase-1 has been shown. To my humble opinion, the manuscript shows scientific soundness. Nevertheless there are some comments to be made.

Minor changes needed in language and grammar. Strongly advise to be reviewed by an English Editor.

The protocol of EV-isolation has nicely been described. It is a well-established methode, butt what I would like to see if the expected EV's are actually present in the pellets. Are there any Electron Microscopy pictures that prove the presence of EV's? This is an important issue, just to be sure that the EVs are responsible for communication, rather than any free proteins.

The authors should describe a bit more in detail how EVs are accepted and incorporated by host cells. this could be done in the introduction or in the discussion. This to clarify this process

Only one cell line has been used to derive EVs from. It is well known that PCa is a heterogeneous disease. Therefore I would like to encourage the authors to deliberate on this. What is known of the effect other EVs from other PCa cell lines? Have you tested other cell lines?

In correlation to nr 4: have other recipient cells been tested like fibroblasts. Probably they are more important in creating a tumor-preferable micro-environment. I would like encourage the authors to describe this in their manuscript. Or: maybe the authors could describe more precisely why they used another prostate epithelial cell line (the PNT2 cell line).   

Are results presented in the manuscript also applicable in other cell lines? Does the same effect occur in other cancerous diseases?

Author Response

Reviewer 2

We greatly thank the reviewer for his/her interesting comments. We added a new experiment in Fig.1 that, in our opinion, improves the MS. We have also amended the MS according to reviewer’s suggestions and hope that it is now suitable for publication.

The authors describe the role of prostate cancer EV's in their micro-environment (and macrophages) very nicely. The role of IL-1B, Caspase-1 has been shown. To my humble opinion, the manuscript shows scientific soundness. Nevertheless there are some comments to be made.

Minor changes needed in language and grammar. Strongly advise to be reviewed by an English Editor.

The protocol of EV-isolation has nicely been described. It is a well-established methode, butt what I would like to see if the expected EV's are actually present in the pellets. Are there any Electron Microscopy pictures that prove the presence of EV's? This is an important issue, just to be sure that the EVs are responsible for communication, rather than any free proteins.

The authors should describe a bit more in detail how EVs are accepted and incorporated by host cells. this could be done in the introduction or in the discussion. This to clarify this process

Response

To address both these points, we added a new panel to fig 1. We showed that round shaped vesicles, stained with a lipophilic dye, adhere to THP-1 cell membrane and are then up-taken. After a 1h exposure, EVs seem to be localized in the multivesicular bodies (Pag. 5 lines205-208).

This result has been discussed at pag. 13, lines 384-391

Only one cell line has been used to derive EVs from. It is well known that PCa is a heterogeneous disease. Therefore I would like to encourage the authors to deliberate on this. What is known of the effect other EVs from other PCa cell lines? Have you tested other cell lines?

In correlation to nr 4: have other recipient cells been tested like fibroblasts. Probably they are more important in creating a tumor-preferable micro-environment. I would like encourage the authors to describe this in their manuscript. Or: maybe the authors could describe more precisely why they used another prostate epithelial cell line (the PNT2 cell line).   

Are results presented in the manuscript also applicable in other cell lines? Does the same effect occur in other cancerous diseases?

Response

We thank the reviewer for these stimulating comments. The addressing of these observations has greatly improved the MS. In particular: we have tested the effect of PNT-derived EVs (Fig. 3E) and showed that they do not have the ability to activate caspase-1. In the revised form of the MS we added deliberation on EVs secreted from other prostate cancer cell lines. See Introduction pag. 3 lines 82-85, discussion pag. 13 lines 379-391 and 403-405. Moreover, we stressed the implication of prostate cancer heterogeneity in the last part of the discussion section. Pag.14 lines 443-451.

Reviewer 3 Report

This is a potentially interesting contribution dealing with the induction of an inflammatory phenotype in non-cancerous prostate cells and a TAM-like polarization in immune cells by extracellular vesicles (EVs) derived from advanced stage prostate cancer cells.

My comments are the following:
1. First, some formal issues. a)  Please scrutinize the reference list. At present, it seems like there is no absolute correspondence between reference denotations in running text and their appearance in reference list. It seems to have occurred a shift (regarding one reference) from about the first ten references in reference list (reference 58 is incomplete).b) Authors speak of "tumor microenvironment". Is there any difference to the more established "tumor stroma"? c) There is a lot of abbreviations in running text, some of them are interpreted but far from all of them. Please give a list of all abbreviations used!

2. p2, line 71(and p10, lines 318-321). Extracellular microenvironment. as it has been discussed in the present paper, concerns in vitro conditions.  The existence of EVs/prostasomes  in microenvironment in vivo, derived from highly malignant prostate cancer metastases is however a reality (see The Prostate, 61:291-297 (2004))

3. p2, line 92. The authors add (extracellularly) a fairly high dose of ATP (5 mM) to cells. Please motivate the addition of ATP!

4. p3, line 94 (and p 8, lines 273-274). Addition of 50 mM KCl to medium means (KCl is supposedly dissociated to almost 100%) a transition from isotonicity (about 300 mOsmol/L) to hypertonicity (about 400 mOsmol/L). Did the authors make any compensation for that or otherwise, please discuss possible consequences of this transition.

5. p3, paragraph starting on line 105. The authors did not include any size exclusion chromatography in isolation procedure of EVs. Please explain! 

6. p11, lines 358-359. Please elaborate the statement: "PC3-EVs exposure caused an increase in phosphorylation" (protein kinase associated to EVs, reference!)                     

Author Response

Reviewer 3

We greatly thank the reviewer for the interesting observations. We amended the MS according to reviewer’s suggestions and hope that it is now suitable for publication

This is a potentially interesting contribution dealing with the induction of an inflammatory phenotype in non-cancerous prostate cells and a TAM-like polarization in immune cells by extracellular vesicles (EVs) derived from advanced stage prostate cancer cells.

My comments are the following:
1. First, some formal issues. a)  Please scrutinize the reference list. At present, it seems like there is no absolute correspondence between reference denotations in running text and their appearance in reference list. It seems to have occurred a shift (regarding one reference) from about the first ten references in reference list (reference 58 is incomplete).

Response

We would like to thank the reviewer for the deep revision of the manuscript and apologise for the mistake in the reference list. The reference section has been amended.

b) Authors speak of "tumor microenvironment". Is there any difference to the more established "tumor stroma"?

Response

Although we are aware that microenvironment comprises stromal cells, because analysing the effects of EVs secreted into cell medium, the term microenvironment seemed to the authors more appropriate. This concept has been added in introduction pag 1, lines 35-39

c) There is a lot of abbreviations in running text, some of them are interpreted but far from all of them. Please give a list of all abbreviations used!

Response

An Abbreviations section has been added

p2, line 71(and p10, lines 318-321). Extracellular microenvironment. as it has been discussed in the present paper, concerns in vitro conditions.  The existence of EVs/prostasomes  in microenvironment in vivo, derived from highly malignant prostate cancer metastases is however a reality (see The Prostate, 61:291-297 (2004))

Response

This point has been added in discussion section pag. 13 Lines 388-390

p2, line 92. The authors add (extracellularly) a fairly high dose of ATP (5 mM) to cells. Please motivate the addition of ATP!

Response

As reported in introduction (pag 2 lines 63-64) the addition of ATP should activate P2X7 receptors, one of the possible second steps needed for inflammasome activation. We choose a 5mM concentration based on previous results obtained by our group cited in material and methods section of this revised version of the MS.

p3, line 94 (and p 8, lines 273-274). Addition of 50 mM KCl to medium means (KCl is supposedly dissociated to almost 100%) a transition from isotonicity (about 300 mOsmol/L) to hypertonicity (about 400 mOsmol/L). Did the authors make any compensation for that or otherwise, please discuss possible consequences of this transition.

Response

As stated in material and methods (section 2.2), we performed a cell viability test to ascertain that the addition of KCl to the medium does not cause any toxic effect due to changes in osmolarity. This concept is now strengthened. Pag. 3 Lines 108-110.

p3, paragraph starting on line 105. The authors did not include any size exclusion chromatography in isolation procedure of EVs. Please explain! 

Response

In pilot experiments we performed a sephadex G200 chromatographic separation that only resulted in a lower yield of EVs without affecting their cellular efficacy. Moreover, we isolated EVs from an untreated, healthy, cell culture, thus we do not expect to have gross material that can significantly affect EVs effects. For these reasons we preferred to isolate extracellular vesicles only by differential centrifugation steps.

p11, lines 358-359. Please elaborate the statement: "PC3-EVs exposure caused an increase in phosphorylation" (protein kinase associated to EVs, reference!)  

Response

The statement has been rephrased. Pag. 14 lines. 424-426.          

Round 2

Reviewer 1 Report

No further comments necessary.